# Genotypes, Enterotoxin Gene Profiles, and Antimicrobial Resistance of *Staphylococcus aureus* Associated with Foodborne Outbreaks in Hangzhou, China

**DOI:** 10.3390/toxins11060307

**Published:** 2019-05-29

**Authors:** Qi Chen, Sangma Xie

**Affiliations:** 1Department of Microbiology, Hangzhou Center for Disease Control and Prevention, Hangzhou 310021, China; 2College of Life Information Science and Instrument Engineering, Hangzhou Dianzi University, Hangzhou 310018, China; smxie@hdu.edu.cn

**Keywords:** staphylococcal food poisoning, staphylococcal enterotoxins, multilocus sequence typing, *spa* typing, pulse-field gel electrophoresis, drug resistance

## Abstract

Staphylococcal food poisoning is an illness caused by the consumption of food that contains sufficient amounts of one or more enterotoxins. In the present study, a total of 37 *S. aureus* isolates were recovered from leftover food, swabs from a kitchen environment, and patient feces associated with four foodborne outbreaks that occurred in Hangzhou, southeast China, and were characterized by multilocus sequence typing (MLST), *spa* typing, pulse-field gel electrophoresis (PFGE), and antimicrobial susceptibility. Classical enterotoxin and enterotoxin-like genes were profiled by PCR analysis. ST6-t304 was the most common clone (40.54%), followed by ST2315-t11687 (32.43%). Six clusters (A to F) were divided based on PFGE patterns, and Clusters A and C were the most common types, constituting 86.49% of all isolates. Moreover, *sea* was the most frequently identified enterotoxin gene (81.08%), followed by the combination of *seg–sei–selm–seln–sleo–selu* and *sec–sell* (each 48.65%). Five isolates also harbored the exotoxin cluster *sed–selj–ser*. In addition, resistance to penicillin (97.30%), erythromycin (37.85), tetracycline (32.43%), clindamycin, gentamicin, and sulfamethoxazole (each 10.81%) was observed. Our research demonstrated the link between leftover foods and patients by molecular typing and detecting the profiles of enterotoxin or enterotoxin-like genes in human and food isolates. *S. aureus* maintains an extensive repertoire of enterotoxins and drug resistance genes that could cause potential health threats to consumers.

## 1. Introduction

*Staphylococcus aureus* is a well-known bacterial pathogen that causes a wide range of clinical infections from superficial skin and soft tissue infections to life-threatening septicemia [1]. *S. aureus* is also a significant cause of foodborne outbreaks, leading to an estimated 241,000 illnesses every year in the United States [2]. Staphylococcal food poisoning (SFP) was the third most frequent bacterial etiology of outbreaks in China after *Vibro parahaemolyticus* and *Salmonella* from 2011 to 2016 [3]. A total of 314 outbreaks of SFP were reported in China, involving 5196 illnesses and leading to one death between these six years. However, the true incidence of SFP could be underestimated due to the nontypical symptoms [4]. The prevalence of this disease makes its control important.

SFP is associated with enterotoxins (SEs) that are synthesized by *S. aureus* during food preparation and processing. SEs are highly stable and highly heat resistant. They are also resistant to proteolytic enzymes and low pH, enabling them to be fully functional in the gastrointestinal tract [5]. So far, 23 distinct SEs have been designated based on their antigenicity (SEA to SElY) [6]. Among them, five classical SEs (SEA, SEB, SEC, SED, and SEE) and more recently described nonclassical SEs, including SEG, SHE, SEI, SER, SES, and SET, exhibit emetic activity [7]. Related toxins that lack emetic activity or have not been tested for emetic activity are designated similarly as staphylococcal enterotoxin-like toxins (SEls) [8,9].

The emergence of methicillin-resistant *S. aureus* (MRSA) from livestock and companion animals [10,11] and the subsequent detection of enterotoxins produced by those isolates highlight the necessity of monitoring *S. aureus* clones from foodborne disease [12,13,14,15,16,17]. The aim of this study was to characterize *S. aureus* isolated from foodborne outbreaks in Hangzhou, southeast China, by testing a capacity for carrying genes encoding SEs, evaluating resistance properties, and determining pulse-field gel electrophoresis (PFGE), MLST, and *spa* typing.

## 2. Results

### 2.1. Phenotypic Characterization and Identification of S. aureus Isolates

*S. aureus* isolates were isolated from various samples associated with four foodborne outbreaks: 14 fecal samples, 22 leftover items, and one cotton swab of a kitchen environment. The amount of *S. aureus* in food remnants was at least 10^5^ CFU/g. Preliminary estimation was carried out by culture properties on a BP plate. Hemolytic activities were determined on sheep blood agar. Furthermore, a 278 bp (*nuc* gene) band detectable in a 1.5% agarose gel was observed in the PCR products of all 37 *S. aureus* isolates.

### 2.2. Antibiotic Resistance

All 37 stains from the food poisoning samples were resistant to one or more kinds of antibiotic. Penicillin (PEN) resistance was the most commonly observed resistance among the tested isolates except for one, followed by erythromycin resistance (37.84%). A total of 29.73% of the isolates (11/37) were only resistant to penicillin, followed by the combination of erythromycin–clindamycin–gentamicin–sulfamethoxazole–penicillin (4/37, 10.81%).

### 2.3. Molecular Genotyping of S. aureus

To achieve more reliable genotyping results and enhance the convenience of comparing different studies, we performed MLST, *spa* typing, and PFGE on the 37 isolates. The allelic profile of seven housekeeping genes revealed eight sequence types (STs), namely, ST5, ST6, ST72, ST188, ST573, ST2250, ST2315, and ST3055 (Table 1). ST6 was the most common ST and was identified in 15 (40.54%) of the isolates from 2 individual outbreaks. Twelve isolates (32.43%) from only 1 outbreak were identified as ST2315. *Spa* typing of all of the isolates yielded nine kinds (Table 1). The top two were t304 and t11687, the corresponding partners of ST6 and ST2315, respectively. It is interesting that isolates with the same ST (ST5) could be divided into two different *spa* types. The 37 isolates were also typed by PFGE (Figure 1). Using an 80% similarity cut-off, the strains were grouped into six clusters, designated by the letters A to F. Clusters A and C were the dominant types, constituting 86.49% of all isolates. Two individual strains in Outbreak 1 exhibited the same ST and *spa* type (ST188-t189) but presented different PFGE types (D1 and D2) with 85.7% similarity. ST6 was also identified in Outbreaks 3 and 4, with the same *spa* type (t304) and almost 100% similarity to the PFGE type. All ST5 strains were classified as Cluster A but with two different *spa* types (t1228 and t002).

### 2.4. Prevalence of Enterotoxin or Enterotoxin-Like Genes

As shown in Table 2, the majority of the isolates in this study contained *sea* gene (81.08%); fewer isolates carried the combination of *seg*–*sei*–*selm*–*seln*–*selo*–*selu* and *sec*–*sell* (each 48.65%). Only five isolates harbored the exotoxin cluster *sed*–*selj*–*ser*. None of the isolates tested positive for the *see*, *selk*, *selp,* and *selq* genes. The number of enterotoxin genes ranged from 1 to 11.

## 3. Discussion

SFP is one of the most prevalent causes of foodborne illness worldwide. Genetic information of those isolates typically differs between countries and even between cities in the same country. In the current study, 37 foodborne-related *S. aureus* isolates from Hangzhou, China, were assigned to 12 strains by molecular typing techniques. MLST provided a comparison of the nucleotide sequences of seven housekeeping genes. MLST has been widely used for epidemic analyses of *S. aureus* related to hospital infections and foodborne diseases since the first report of MLST use for *S. aureus* [18]. In South Korea, ST1, ST59, and ST30 strains were the clones that were most frequently associated with SFP [19]. ST45 and ST81 were more closely related to SFP outbreaks in Japan [20,21,22]. ST45 and ST5 have been identified in SFP outbreaks in Europe [23,24]. In this study, *S. aureus* ST6-t304 was the predominant lineage isolated from two separate outbreaks. Previous SFP reports occurred in Shenzhen, Ma’anshan, and Xi’an, and it was demonstrated that ST6-t701 was the most common type in China. It is interesting that ST6-t304 is the ancestor of ST6-t701, which had one *spa* repeat (r25) based on BURP analysis results [25,26]. In addition, another two dominant lineages in our study, namely, ST188-t189 and ST5-t002, were reported in other SFP outbreaks and food samples in China [25,26,27,28], suggesting that additional studies are needed to confirm the temporal relationship of these strains.

In this study, PFGE was used in the analysis of the molecular epidemiological diversity of the strains, especially in relation to the analysis of short-term epidemiology due to high discrimination. Based on PFGE analysis, all outbreaks were caused by multiple clones (A to F), which was identical to the results from the MLST methods and *spa* typing. In Outbreaks 1 and 3, two similar PFGE patterns (C1 and C2) were detected with a similarity of 90.3%, whereas the *spa* and MLST types remained indistinguishable. According to the criteria proposed by Tenover et al., a PFGE pattern with a 4–6-band difference can be regarded as possibly related to outbreak strains [29]. Four obvious bands were detected between Strains 1 and 10. A period of approximately 10 months may be the reason for simple insertions or deletions of DNA. Outbreaks 3 and 4 may have been caused by the same epidemiological strain, which was supported by the fact that Strains 9 and 11 had the same MLST, *spa* types, and PFGE patterns. Although these two outbreaks occurred in two different districts, the period between the outbreaks was less than two months. The ST188-t189-D1 clone had the same ST and *spa* type as the ST188-t189-D2 clone, while several bands were detectable between these two strains, indicating some large-scale changes in the accessory genome. This situation was also found in the isolates ST5-t002-A1 and ST5-t002-A2. In Outbreak 2, there were two *spa* types (t002 and t1228) with the same PFGE pattern (A1). According to the *spa* repeat analysis, there was only one repeat difference between these two clones, suggesting that these strains may belong to the same clone and differed only due to the recombination of select genes [25]. Based on the above comparison, PFGE is proposed to be more effective in describing the strain population.

SFP is associated with one or more families of genes that code for heat-stable enterotoxins. This study demonstrated that the *sea* gene was the dominant enterotoxin (81.08%) in *S. aureus* strains recovered in this study. Additionally, *sea* was described to be the most prevalent gene recovered from not only clinical samples but also food- and foodborne-related samples in other studies [25,26,30,31]. Yan et al. reported that SFP caused by *sea* alone may be unique in Shenzhen. Fortunately, Strains 9 and 11 in our study also harbored only *sea* genes. The ST188-t189-D2 clone only had the *seb* gene, which was exactly similar to the strain from the studies conducted by Yan et al. and Song et al. [25,27] in comparison with MLST, *spa* typing, and PFGE bands. Nearly all *se* and *sel* genes are associated with mobile genetic elements, including plasmids, prophages, *S. aureus* pathogenicity island (SaPI), and genomic island *ν*Sa or next to the staphylococcal cassette chromosome (SCC) [23]. In our study, the combination of *seg, sei, selm, seln, selo,* and *selu* was detected at a relatively high frequency, forming an operon encoded by enterotoxin gene cluster 2 (*egc2*) in the *v*Sa*β* genomic island inserted at specific loci in the chromosome [23]. Genotype *sec–sell* and *sed–selj–selr* could be a combination of Type II *v*Sa3 and pIB485, respectively, which was also observed in this study. Strains 2 and 4 or Strains 7 and 8 harbored the same MLST and *spa* type and almost the same PFGE bands, but the distribution of *se*/*sel* genes was largely distinct. This finding suggested that toxin gene transfer between *S. aureus* plays a considerable role in genetic diversity [7].

The majority of the isolates in this study were resistant to one or two antibiotics. The high rate of resistance to penicillin was in agreement with previous studies of *S. aureus* isolates from food samples or foodborne outbreaks in both China and other countries [25,26,32,33]. There was only one base difference in *spa* sequence among ST5-t002 and ST5-t1228, which were found to be multidrug-resistant to five antibiotics. ST5-t002 in the studies conducted by Li et al. and Argudin et al. was also identified as resistant to at least three drugs [25,26], indicating that a deep investigation may be needed to explain the relationship between genomic characterization and multidrug resistance.

## 4. Conclusions

Our study demonstrated the SE/SEl profiles and genetic diversity of *S. aureus* isolated from foodborne outbreaks in Hangzhou, China. The detection of not only classical SEs but also new SE/SEls should be considered for the diagnostic analysis of SFP outbreaks due to the frequency of their observation in our study. Molecular genotyping methods, including MLST, PFGE, and *spa* typing, were useful for investigating epidemiological relatedness and even contaminated food sources, especially for outbreaks occurring in the same city.

## 5. Materials and Methods 

### 5.1. Bacterial Isolates

Isolates were collected from the fecal specimens of individual patients, swabs from the environment, and food samples associated with four foodborne outbreaks in Hangzhou, Zhengjiang Province, China, during 2015–2016. A total of 37 *S. aureus* isolates were identified according to the methods described in GB 4789.10-2010 Food Microbiological Examination: *S. aureus*. In general, test isolates were cultured in tryptic soy broth, Baird-Parker agar, BHI, and tryptic soy agar (TSA). The suspected single clones were then tested by Gram staining, coagulase testing, hemolytic activity (determined on sheep blood agar), and catalytic reactions using the Vitek 2 compact system. Furthermore, all isolates were investigated for the presence of the *nuc* gene (*S. aureus* species specific). Bacterial stocks of each isolate were maintained at −80 °C in tryptic soy broth (TSB) containing 20% glycerol (*v*/*v*). All the isolates were thawed and subcultured in TSA for 18–24 h prior to use.

### 5.2. Antimicrobial Susceptibility

Susceptibility to antimicrobial agents was tested using commercially available plates (Scenker, Liaocheng, China) that contained oxacillin (OX, resistance breakpoint ≥4 μg/mL), erythromycin (ERY, ≥8 μg/mL), clindamycin (CD, ≥4 μg/mL), levofloxacin (LEV, ≥4 μg/mL), tetracycline (TE, ≥16 μg/mL), gentamicin (GEN, ≥16 μg/mL), vancomycin (VA, ≥16 μg/mL), teicoplanin (TEC, ≥32 μg/mL), rifampicin (RA, ≥4 μg/mL), sulfamethoxazole (SXT, ≥4/76 μg/mL), daptomycin (DAP, >1 μg/mL), penicillin (PEN, ≥32 μg/mL), and cefoxitin (FOX, ≥25 μg/mL). *S. aureus* ATCC 29213 was included as a quality control strain in the study. 

### 5.3. Extraction of Genomic DNA

*S. aureus* isolates were grown overnight at 37 °C in BHI broth. Genomic DNA was extracted using a bacterial genomic DNA extraction kit (DNeasy Blood and Tissue Kit, Qiagen Inc., Redwood City, CA, USA) according to the manufacturer’s instructions.

### 5.4. Detection of 23 SE/SEl Genes

The primer sets used to detect the *sea*, *seb*, *sec*, *sed,* and *see* genes were those described by Becker et al. [34], *seg*, *seh*, *sei*, *selj*, *selk*, *selm*, *seln,* and *selu* were those described by Tang et al. [31], *selo*, *selp*, *selq*, *sell,* and *ser* were those described by Omoe et al. [35]. PCR was performed using a commercial PCR kit (Takara, Kusatsu, Japan) in a 25 μL volume containing 2.5 μL of buffer, 2 μL of dNTP (2.5 mM), 0.125 μL of Taq DNA polymerase (1 U/μL), 0.5 μL of each primer set (10 μM), and 1 μL of template DNA. PCR was performed on a Mycyler thermal cycler (Bio-Rad, Hercules, CA, USA) using the following steps: initial denaturation at 95 °C for 5 min, 30 cycles of 95 °C for 30 s, each annealing temperature for 30 s and 72 °C for 45 s, and a final extension at 72 °C for 5 min. The PCR products were separated with electrophoresis in 1.5% agarose gel to verify the expected size of the amplicons.

### 5.5. PFGE

PFGE of the *S. aureus* isolates was performed in a CHEF Mapper system (Bio-Rad Laboratories, Hercules, CA, USA) as described previously [36] with some modifications. In this study, DNA from *Salmonella choleraesuis* serotype Branderup H9812 digested with *XbaI* (New England Biolabs Inc., Ipswich, MA, USA) was included as a molecular size marker. The banding patterns were analyzed with Denmark BioNumerics, version 6.6 (Applied Maths, Sint-Martens-Laterm, Belgium) with a 1% optimization and a band-matching tolerance of 1%.

### 5.6. MLST and Spa Typing

Genomic DNA, extracted as indicated above, was used in PCR amplifications for the *spa* type and seven housekeeping genes (*arcC*, *aroE*, *glpF*, *gmk*, *pta*, *tpi,* and *yqiL*) listed on the *S. aureus* MLST website [37]. The allelic profile of an S. aureus isolate appearing as an array of seven allele numbers was achieved by sequencing (Shangong, Shanghai, China). The Ridom Spa Server [38] and the MLST website [37] were used to assign *spa* types and sequence types (ST), respectively. 

## Figures and Tables

**Figure 1 toxins-11-00307-f001:**
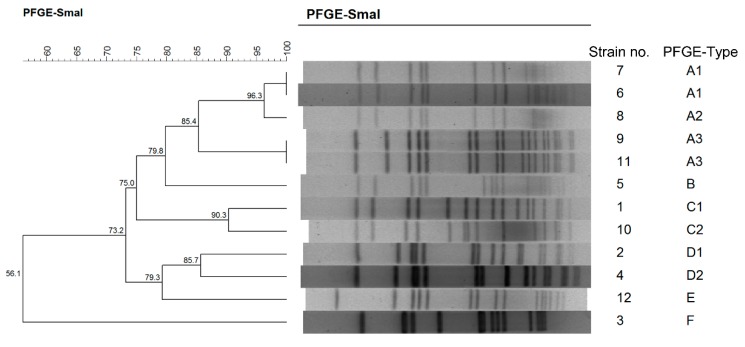
Comparison of *S. aureus* from outbreaks using PFGE.

**Table 1 toxins-11-00307-t001:** Overview of *S. aureus* isolates analyzed in this work.

Outbreaks	Data	District	Origins (no. of Isolates) ^a^	Strains ^b^	MLST	Spa	PFGE	Resistance ^c^
1	October, 2015	Gongshu	Food (8) Environment (1) Feces (3)	1	ST2315	t11687	C1	TE–PEN
			Feces (1)	2	ST188	t189	D1	PEN
			Feces (1)	3	ST2250	t7960	F	PEN
			Feces (1)	4	ST188	t189	D2	PEN
2	June, 2016	Xiangcheng	Food (1)	5	ST3055	t084	B	PEN
			Food (1)	6	ST5	t1228	A1	ERY–CD–GEN–SXT–PEN
			Food (1)	7	ST5	t002	A1	ERY–CD–GEN–SXT–PEN
			Feces (1)					
			Feces (1)	8	ST5	t002	A2	ERY–CD–GEN–SXT–PEN
3	September, 2016	Gongshu	Food (6)	9	ST6	t304	A3	PEN
			Feces (1)					
			Feces (1)	10	ST573	t458	C2	ERY–PEN
4	October, 2016	Xihu	Food (5)	11	ST6	t304	A3	ERY–PEN
			Feces (3)					
			Feces (1)	12	ST72	t148	E	ERY

^a^ A total of 37 *S. aureus* isolates were identified. The origins of the samples included food (22 isolates), feces (14), and swabs from the environment (1). ^b^ The 27 foodborne isolates were assigned to 12 strains *^c^* PEN, penicillin; ERY, erythromycin; TE, tetracycline; CD, clindamycin; GEN, gentamicin; SXT, sulfamethoxazole.

**Table 2 toxins-11-00307-t002:** The profiles of *se/sel* genes in four outbreak-related *S. aureus* strains.

Strains												
SE Genes	1	2	3	4	5	6	7	8	9	10	11	12
*sea*	+	+							+	+	+	+
*seb*				+								
*sec*	+	+				+	+			+		+
*sed*					+	+	+	+				
*see*												
*seg*	+					+	+	+		+		+
*seh*			+									
*sei*	+					+	+	+		+		+
*sej*					+	+	+	+				
*sek*												
*sel*	+	+				+	+			+		+
*sem*	+					+	+	+		+		+
*sen*	+					+	+	+		+		+
*seo*	+					+	+	+		+		+
*sep*												
*seq*												
*ser*					+	+	+	+				
*seu*	+					+	+	+		+		+
Total	9	3	1	1	3	11	11	9	1	9	1	9

+: Genes were detected positive by PCR

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
