# Peer review of "Genotypes, Enterotoxin Gene Profiles, and Antimicrobial Resistance of Staphylococcus aureus Associated with Foodborne Outbreaks in Hangzhou, China"

_toxins, 2019, doi:10.3390/toxins11060307_

Round 1
Reviewer 1 Report
Overall the manuscript is well-written, the research subject is worthy of investigation and fits within the scope of the journal. The experimental design appropriate and results achieved are sound.
Author Response
Thank you for your suggestion.
Reviewer 2 Report
Overall this is a nice manuscript that describes new clinical isolates of antibiotic resistant S. aureus. My only major issue is the use of the word "strain" when many times these should be referred to as "isolates". There is no mention of access to these isolates: have they been deposited in a culture collection?
Also there are several sentences that do not make sense, or convey a wrong message, which could be resolved by some careful editing (also the formatting in table 1 makes it difficult to quickly discern which isolates were from particular outbreaks). Finally, as a reader I would appreciate some information regarding the outbreaks (date, location - perhaps a map to show distance, number of confirmed cases, etc).
Author Response
Point 1: Overall this is a nice manuscript that describes new clinical isolates of antibiotic resistant S. aureus. My only major issue is the use of the word "strain" when many times these should be referred to as "isolates". There is no mention of access to these isolates: have they been deposited in a culture collection?
Response 1: Thank you for your suggestion. We have used the term "isolates" in place of "strains" in line 15, 62, 71, 96, 109, 110, 111, 155, 195, 198. And we have added some information of the isolation of the bacteria in the "Materials and Methods". These isolates have been deposited in a culture collection.
Point 2: Also there are several sentences that do not make sense, or convey a wrong message, which could be resolved by some careful editing (also the formatting in table 1 makes it difficult to quickly discern which isolates were from particular outbreaks).
Response 2: Thank you for your suggestion. We have made some revisions to the Table 1. To avoid any misunderstanding, the information was displayed in the format of regular table in stead of Three Line Table.
Point 3: Finally, as a reader I would appreciate some information regarding the outbreaks (date, location - perhaps a map to show distance, number of confirmed cases, etc).
Response 3: Thank you for your suggestion. Honestly, we had intended to provide more information about the outbreaks before submission, but were put off by the decision of some colleagues. In the CDC, epidemiological investigation and laboratory work about any foodborne outbreaks was charged with two individual departments: the Institute for Communicable Disease Control and Prevention and the Department of Microbial Detection. We have negotiated with the colleagues of the Institute for Communicable Disease Control and Prevention, but failed because they plan to organize the epidemiological data for further publications. The information of the first outbreaks was complete as it had already published on a Chinese magazine. We only could provide the date and the District of another three outbreaks in this publication. Please refer to Table 1 in the updated manuscript.
Reviewer 3 Report
The
manuscript (481070) reported “Genotypes, Enterotoxin Gene Profiles and
Antimicrobial Resistance of Staphylococcus aureus Associated with
Foodborne Outbreaks”. The authors demonstrated the link between leftover
foods and patients by molecular typing and detecting the profiles of
enterotoxin or enterotoxin-like genes in human and food isolates. S. aureusmaintains
an extensive repertoire of enterotoxins and drug resistance genes that
could cause potential health threats to consumers. I considered well the
scientific contributions of the present manuscript and the contents
will be useful for researchers in the author’s country, but I think that
the conclusions of this manuscript are not new and are already well
known to many researchers, because many similar papers have been
reported in many other journals.
Author Response
Thank you for your suggestion.
Round 2
Reviewer 3 Report
Tmanuscript (481070) reported “Genotypes, Enterotoxin Gene Profiles and Antimicrobial Resistance of Staphylococcus aureus Associated with Foodborne Outbreaks”. The authors demonstrated the link between leftover foods and patients by molecular typing and detecting the profiles of enterotoxin or enterotoxin-like genes in human and food isolates. S. aureus maintains an extensive repertoire of enterotoxins and drug resistance genes that could cause potential health threats to consumers. I considered well the scientific contributions of the present manuscript and the contents will be useful for researchers in the author’s country, but I think that the conclusions of this manuscript are not new and are already well
known to many researchers, because many similar papers have been reported in many other journals.
Author Response
Thank you for your suggestion. We had modified the title to "
Genotypes, Enterotoxin Gene Profiles and Antimicrobial Resistance of Staphylococcus aureus Associated with Foodborne Outbreaks in Hangzhou China".